# Enforcement of Fairness Norms by Punishment: A Comparison of Gains and Losses

**DOI:** 10.3390/bs14010039

**Published:** 2024-01-05

**Authors:** Ivo Windrich, Sabrina Kierspel, Thomas Neumann, Roger Berger, Bodo Vogt

**Affiliations:** 1Institute of Sociology, University of Leipzig, 04107 Leipzig, Germany; roger.berger@uni-leipzig.de; 2Empirical Economics, Otto-von-Guericke University Magdeburg, 39106 Magdeburg, Germany; 3University Department of Neurology, Otto-von-Guericke University Magdeburg, 39120 Magdeburg, Germany; 4Health Services Research, School of Life Sciences, University of Siegen, 57076 Siegen, Germany; 5Institute of Social Medicine and Health Systems Research, Otto-von-Guericke University Magdeburg, 39120 Magdeburg, Germany; 6Center for Behavioral Brain Sciences (CBBS), 39106 Magdeburg, Germany

**Keywords:** fairness, social norms, punishment, dictator game, third-party-punishment game, loss domain

## Abstract

Although in everyday life decisions about losses are prevalent (e.g., the climate crisis and the COVID-19 crisis), there is hardly any research on decisions in the loss domain. Therefore, we conducted online experiments with a sample of 672 participants (mostly students), using third-party punishment dictator games (DGs) in the loss domain to explore the impact of losses and punishment threats on the conformity to the fairness norm. Subjects in the treatment condition have to divide a loss of −10 € with the threat of a third-party punishment with different strengths (control: gains, no punishment). Overall, the statistical evidence seems rather weak, but when it comes to losses, subjects are more rational and straightforward with their words and deeds than with gains. Therefore, in the loss domain, subjects are more likely to believe that the fairness norm should be followed, and they subjectively perceive that the others do as well. Furthermore, although dictators’ decisions are more selfish in the loss domain, dictators there react more strongly to the punishment threat by reducing their demands than in the gains domain. This holds as long as the punishment threat is strong enough, as judged from a rational perspective.

## 1. Introduction

Social norms are usually seen as rules of prescribed or proscribed behaviour for certain situations that are socially enforced in some way (e.g., [1]). There are two ideas of how to define and measure a social norm. One idea, supported by Bicchieri [2], is idealistic. Bicchieri uses the term “constructivist” and defines a norm regarding people’s preferences, beliefs, and expectations. The other perspective is behavioristic and defines a norm regarding what people do in a certain situation. This perspective is, among others, the one advocated by Gächter and Fehr [1] and the sociologist Popitz [3,4]. Therefore, a norm is either a rule of behaviour that exists in people’s minds or a behavioural regularity. There are strong arguments for both perspectives. Advocates of the behavioural perspective argue that only behaviour can be observed objectively. Therefore, Popitz [3] writes that norms cannot be elicited with surveys because the concept of norms refers to actual behaviour [4]. Bicchieri [2], meanwhile, argues that some norms influence behaviour by avoiding the prescribed action. Therefore, the prescribed action does not happen, and there is no behavioral regularity to observe. That is why a norm exists as preferences as much as behavioral expectations. We follow both ideas here and measure norms as observed behavior and as surveyed social expectations.

Most empirical research on fairness norms (and, beyond that, on most topics of economics, sociology, and probably any other social science) has been conducted in the gains domain. Therefore, a fairness norm is applied to some gain that has to be split. Nevertheless, we argue that this is only one application. In many social situations, there is nothing to gain but only—more or less—to lose. In such situations, fairness norms concern the division of losses. Just looking at the present (e.g., the climate change problems and the COVID-19 crisis, with losses of health and even lives), one might be apt to say that (normatively guided) decisions about losses are the common case and decisions about gains the exception. Due to such considerations, we conducted research on negative games in which losses are divided [5,6].

In this contribution, we discuss experimental results for the fairness norm in the loss domain with third-party punishment (TPP). The following questions therefore arise: Do people conform with the fairness norm in the division of losses even if it might be costlier than with gains? Does the threat of being punished for not being fair lead to more or less fairness in the loss domain than in the gains domain? These are not trivial questions because on the one hand, the punishment in the loss domain might be additionally painful. However, it might also be for the punisher who has to bear the costs of the punishment. Therefore, on the other hand, people might be more prone to take the risk of being unfair and hope not to be punished.

Fairness norms can be measured using the Dictator Game (DG). For the fairness norm in the loss domain, this happens with the negative DG, in which a loss has to be divided. Therefore, at best, the dictator remains in the status quo without bearing any loss. We are aware that there is an ongoing debate regarding whether the DG elicits a social norm and if so, which norm plays a role [2,7]: “Worries of ecological validity arise in the baseline DG design because of the absence of familiar norms that apply to this situation” [7] (p. 581). For other games, such as the Ultimatum Game and the Public Goods Game, there seems to be a consensus that social norms of fairness or cooperation affect decision-making [2,8,9]. For example, in their cross-cultural studies, Henrich et al. [9] usually interpret experimental results in this way. For the DG, there are several competing interpretations of what is actually measured with the game. The most common interpretation of dictator giving is altruism or benevolence (cf. [10]). However, some authors insist that dictator giving is a methodological artifact resulting from insufficient anonymity [11,12]. In our study, we explain behavior in DGs from a norm-theoretical perspective. This means that there must be expectations from the social environment for the dictator to behave in a certain way, yet it is not clear in which way. It depends on how the actors categorize the DG [2,3]. If the task in the DG is interpreted as giving, then a norm of “right giving” will be activated, saying that the dictator should give at least something. If the dictator interprets the task as sharing, then the social norm requires them to share equally.

How can we discover whether there is a social norm active in DGs, and if so, which one? Per the idealistic perspective on norms [2], we should ask participants which behavior they consider appropriate or inappropriate. Per the behavioral perspective on norms [4], we would examine behavioral regularities. A behavioral regularity, then, is a social norm as soon as a deviation from it is punished by others. Considering what is typical in the DG, most researchers would agree that positive giving occurs [10], but there is no consensus regarding whether there is an equality norm active in the game. Some authors [13,14] argue that a bimodal distribution of offers typically occurs in the DG. In our DG experiments, the behavioral regularity is doubtlessly the equal split [6].

A second question arises from the behavioral perspective of norm theory: Is there punishment for deviant behavior? One could interpret the Ultimatum Game as a DG with second-party punishment. We know for the Ultimatum Game that the equal split is a behavioral regularity and that the responder sanctions low offers [10]. However, to be sure that there is a social demand for equal splits in the DG, we should investigate whether a third person who is not involved in the DG’s outcome would sanction low offers. Therefore, we conducted TPP-DGs to determine whether there is a relevant proportion of punishment of unfair others and whether offers become fairer. Punishment behavior is strong empirical evidence that there are social expectations regarding the dictator’s behavior, indicating that there is indeed a social norm of fair giving/sharing active in the game.

### 1.1. Research Questions

Sanctioning of norm deviance typically is costly, which is why norm-driven punishment is often called “altruistic”. We already know that altruistic punishment occurs in the Public Goods Game [15]. From TPP-DGs [16,17], we know altruistic punishment of low offers occurs in the DG.

Our research question is twofold. First, we investigate whether there is a different impact of introducing a punishment mechanism in the DG when the game is played involving losses compared to the standard version involving gains. Second, we varied the punishment mechanism’s strength to see whether there is a different impact of the punishment mechanism’s strength on fairness behavior. Recent studies [18] have shown that the strength of punitive power affects cooperative behavior in Public Goods Games.

On both points, we would like to give some explanations. First, we had conducted experiments on this question for Dictator and Ultimatum Games involving losses [5,19,20] and on norm-focusing treatments in a DG involving losses [6], in all cases compared to behavior in games involving gains. For several treatments, we usually found that offers in the loss domain are more selfish than in the gains domain. However, the effect is rather weak and the difference not always statistically significant. Other studies on the difference in fairness behavior between gains and losses yielded mixed results. There is only one general result: whether gains or losses are to be divided affects fairness. Studies [21,22] have confirmed our finding that dictators become more self-interested when dividing losses compared to gains. Researchers [23,24,25] have made the opposite observation, that dictators in the loss domain became more fair.

From a theoretical perspective, there are also arguments for differences in fairness between the loss and gains domains. Therefore, the theories of loss aversion [26] and loss attention [27] would predict different fairness behavior for a negative DG compared to a DG involving gains. The argument regarding loss aversion [26] refers to a stronger evaluation of a loss than of a gain of the same absolute amount. When playing fair costs a dictator the same amount of money in positive and negative DGs, but the evaluation of this amount is increased in the loss domain, then playing fair will be costlier in terms of the value of the amount to share when one shares a loss than when one shares a gain. A behavior that is costlier will be executed less often. Therefore, dictators in the loss domain should play less fairly than dictators in the gains domain. Loss attention theory [27] claims that a situation of losses increases attention on the decision. This might lead to a more rational decision. Dictators with a loss to divide might think more about whether they want to conform to a fairness norm or deviate rather than simply automatically deciding in accordance with the norm [2].

Second, for social norms, a conflict always arises between norm adherence and self-interested behavior [2]. Therefore, with fairness norms, such as those in the Ultimatum Game and DG, there is always a material incentive to play unfairly and maximize one’s own outcome. This is one reason many norm theorists [4,28] emphasize that social sanctioning of norm deviance is crucial for the maintenance of the norm. From a rational-choice perspective, the punishment mechanism needs to be strong enough to enforce conformity [28]. This is the case as soon as the costs of getting punished are greater than the possible gain from norm deviance. In the real world, sanctioning behavior varies extensively. It ranges from the rejection of being kind to a person or avoiding a personal interaction to sending criminals to prison. Therefore, we explored the impact of various strengths of a punishment mechanism in the DG with TPP.

Several TPP experiments have shown that there is a meaningful amount of altruistic punishment [16,17]. Fehr and Gächter [15] found that introducing a punishment mechanism in a Public Goods Game leads to a distinct increase in cooperation. Balliet et al. [29] found that both reward and punishment by third parties has a substantial positive effect on cooperation. For the DG with TPP, the study by Fehr and Fischbacher [16] is ground-breaking. They played Prisoner’s Dilemma (PD) and DG with TPP and found that there is a high degree of altruistic punishment for violating cooperation and fairness norms. However, for the DG, they did not find an effect of TPP on conformity to a fairness norm. The level of fairness dictators exhibited was “quite similar to that observed in typical DGs without punishment” [16] (p. 69).

We base our experimental design on the study by Fehr and Fischbacher [16], but there were some methodical differences. First, we use the direct response method instead of the strategy method for third-party decisions because we are mainly interested in the effect of punishment on fairness and there might be differences in behavior between “hot” and “cold” punishment [30]. Second, for reasons of practicability, in the loss treatment, we endow our third parties with the same amount (10 EUR) as the dictator. Fehr and Fischbacher [16] endow the punishers with 50% of the amount to be shared by the dictator. This means our third parties are always advantaged, making punishment less costly relative to the dictator’s possible outcome. Third and most important, we varied the ratio of the punishment costs for the punisher to punishment’s impact on the dictator. In the literature, this ratio is usually 1:3 [15,16,17,31,32]. We think this ratio is quite strong, and in the real world, there are weaker relationships. That is why, in addition to this strong ratio, we introduce a moderate mechanism with a ratio of 1:1 and a weak mechanism with a ratio of 2:1. Furthermore, we hold the costs of punishment constant at EUR 2 for the punisher, so the punisher always had only one decision: punish or not.

Since the initial study on third-party punishment, by Fehr and Fischbacher [16], many studies have been conducted on this topic. In most studies on TPP in DGs, the researchers are interested in the driver of punishment behavior [33,34]. For the DG, it is known that introducing “moral wiggle room” for the dictator decreases fairness [35]. Stüber [36] investigates the question of whether moral wiggle room for the third party impacts behavior. Interestingly, it decreases punishment behavior but not the dictator’s fairness decisions.

To our knowledge, in only two studies similar to ours do the researchers vary the punishment mechanism’s strength: Kamei [32] and Falk et al. [37]. Both played a PD with punishment. Kamei [32] played a PD with TPP and varied the punishment mechanism’s strength by changing the group size of the punishers. He found that norm conformity (cooperation) in PD increases with a larger group of third parties, indicating that a larger group of punishers evokes a stronger punishment mechanism. Falk et al. [37] played a three-person PD with second-party punishment, in which each player could sanction the other two. They compared a low- and a high-sanction condition and found differences in sanctioning behavior, but they did not investigate the effect of different punishment conditions on conformity to a cooperation norm in PD.

We are aware of only one study in which a TPP game was played in the loss domain. Liu et al. [38] played a DG with TPP in the loss context and found that both compensation and punishment by third parties were stronger in the loss domain than in the gains domain. Because they did not really play the DG but showed fictive results to the third parties, they did not investigate the effect of TPP on fairness in decisions on losses. Therefore, our study is the first on how different punishment mechanisms affect fairness behavior in the DG involving losses.

### 1.2. Hypotheses

We use two theoretical perspectives on the question of punishment on norm deviance: a norm-theoretical perspective and a rational-choice view. From the norm perspective, we assume that people usually want to follow given norms because it means that they are socially integrated. Bicchieri [2] defines social norms and says that a norm exists in a population when there is a sufficiently large subpopulation of “conditional followers”. These actors prefer to obey the given rule because they think that the prescribed behavior is reasonable. However, they conform to the rule only on the condition that others follow the rules, too, and that subjectively, they feel obliged that others expect conformity from them.

For people with the (conditional) preference to follow a specific norm (e.g., to cooperate or to play fair), punishment is seen as negative because it shows that their behavior is socially disapproved. From this social-psychological view, the cost of being punished is not so much the material cost but the psychological effect of feeling guilty for behaving in a socially disapproved manner. The punishment’s strength, then, is a signal of the social disapproval’s strength. It follows that people who want to follow the rules avoid punishment and the stronger the punishment mechanism, the more they adhere to a social norm.

In contrast, from a rational-choice perspective, people having self-interested preferences will avoid norm-deviant behavior because the (here, material) costs of being punished reduces their own outcome. However, this is only evident if the punishment mechanism is sufficiently strong [28]. If the material costs of being punished are too low, they are simply factored in without changing the behavior. From this perspective, only a strong enough punishment can increase norm conformity. For a weak punishment, there might even be a counter effect because the rational self-interested dictator includes the costs of being punished in their calculation. Therefore, if they would give a least a small amount, this would not happen anymore because it is offset by the expected punishment.

Therefore, our hypotheses about the impact of punishment and the strength are as follows:

**H1:** ***Effect of punishment mechanism**: The integration of a third-party-punishment mechanism into a DG should increase fairness behavior so that dictator’s demands decrease on average*.

**H2:** ***Effect of strength of punishment mechanism**: The greater the impact of the punishment on the dictator, the more dictators will be fair*.

**H3:** ***Effect of weak punishment mechanism**: Self-interested dictators will offset the expected costs of a punishment in their dictator demands. This leads to increased demands and less fairness for a weak punishment mechanism*.

In our experiments, all punishment mechanisms are costly for the punisher; only the ratio of punishment costs (punisher) to punishment impact (dictator) varies across treatments. From a strict game theoretical point of view in which the dictator has self-interested rational preferences and expects self-interested rational preferences from the punisher, the game can be solved through backward induction. Because punishment is costly for the punisher, there should be no punishment at all for the punishers’ self-interested preferences. The dictator’s expectation that there will be no punishment should lead them to play the game like the standard DG. Therefore, the baseline prediction from the standard rational-choice perspective is that no punishment mechanism will affect behavior at all.

We assume that social norms are the main drivers of behavior in standard experimental games. For the DG, there is a norm of fair sharing involved, which drives dictator giving [6]. This social norm of fairness leads dictators to give appropriate amounts to the receiver. In a TPP-DG, this social norm of fairness will lead to sanctions of “unfair” offers. The punisher who has internalized a social norm of fair sharing will bear some costs to punish low offers in order to enforce this social norm [16,17].

We expect different behavior in the DG involving losses compared to those involving gains. Per prospect theory [26,39], we would expect greater demands in the loss domain. If dictators are loss averse, obeying a fairness norm becomes costlier in the loss domain. In our control experiments [5], we found that the dictator’s demands are greater in the loss domain. The question now is what will happen when we introduce a punishment mechanism in the game? A baseline prediction is that the punishment mechanism does not affect the difference between gains and losses. In our population, we found dictators to be less fair when dividing losses than when they divide gains. Therefore, we can expect this also to happen when introducing a TPP mechanism in DGs involving losses and gains.

On the other hand, there might be an interaction effect of punishment mechanism and loss treatment. It can be argued that loss aversion [26] leads to an increased fear of being sanctioned for unfair behavior because these costs are evaluated more extensively. However, at the same time, according to prospect theory [26,39], actors are risk loving in the loss domain. This might push dictators to take the risk of being sanctioned and possibly avoid costly losses. Theoretically, it remains unclear whether this would just balance more highly evaluated costs or overcompensate for them. For simplicity’s sake, we assume the first. Alternatively, loss attention [27] might increase the attention on the possibility of being sanctioned and bear additional costs for the dictator. Therefore, if the punishment mechanism affects fairness behavior, it might be even stronger in the loss domain. Therefore, overall, we have two competing hypotheses for punishment in a negative TPP-DG.

**H4:** ***No effect of punishment on the difference between gains and losses**: The baseline prediction is that the same pattern will emerge between gains and losses for games, compared to games without a punishment mechanism. In our population, dictator demands were higher in the loss domain. Therefore, we will find greater dictator’s demands in the loss domain for the DG with third-party punishing, too*.

**H5:** ***Interaction effect of punishment mechanism and losses**: Due to loss aversion and/or loss attention, the fear of being sanctioned might increase in the loss domain. Therefore, the effect on fairness by introducing a punishment mechanism to the DG will be stronger for the DG involving losses than for that involving gains*.

## 2. Materials and Methods

Our experiments combined two treatments of the DG. First, we compared DG on losses to DG on gains. These two conditions form our control design to explore the effects of punishment on the dictator. Second, we conducted the DG with TPP in both the gains and loss domains. The punishment mechanism for the dictator was simple. If they received punishment, their outcome was reduced by a certain amount. This amount varied over three conditions, generating a weak, a moderate, and a strong punishment condition. Thus, overall, we have a 2 × 4 design, as shown in Table 1. All games were one-shot plays with random matching of participants in a between-subject design. Next, we explain the design of the experiments in more detail. Data are available in the Appendix A.

### 2.1. Design of Control Experiments

As control treatments, we ran DG experiments without punishment in the gains and loss domains. These experiments were run in 2018 (in the lab) [5] and 2021 (online) [40], and they serve as reference points to measure the effect of punishment on fairness decisions. In these experiments, we conducted Dictator and Ultimatum Games and randomized the order of play. In the current study, we used only DG results in which the DG was played first. The amount to share was always EUR 10.

In the gains condition, the dictator divided EUR 10 with an anonymous receiver in an increment of 50 cents. Subjects were randomly paired, and the roles of dictator or receiver were also assigned randomly. All participants received the same instructions. We used “Player 1” for the dictator and “Player 2” for the receiver. Participants were told that “Player 1 receives an amount of EUR 10, which they have to divide between themselves and Player 2”. Player 1 made a decision by “choosing their own share” of the EUR 10 on a number line. The remaining amount for Player 2 was also shown to Player 1. Player 2, as receiver in the DG, did not make any choice, only being informed about Player 1′s decision. Both players were paid according to the decision of Player 1. In addition, subjects received EUR 5 for their participation.

In the loss condition, we used a prepaid mechanism [41] to induce losses. In this procedure, the experiment consisted of two sessions. Participants were always invited for both sessions, and during enrollment, they received a registration confirmation for both sessions. In the first session, subjects filled out a Big Five survey and received money for participating in the experiment. The amount received consisted of a EUR 5 show-up fee and the EUR 10 for playing the DG. The subjects were told only that they would receive an endowment, and that this endowment could be reduced in the second session. Subjects signed a form indicating that they understood this condition and that they agreed to participate in the second session. The second session took place two weeks later at the same time of day. In the second session, the DG on losses was played. Instructions were kept analogous to the gains treatment, with the exception that a gain of EUR 10 was not endowed to Player 1, but rather a loss of EUR −10. Player 1 was asked to divide this loss with Player 2, and both players were told that they had to pay back the loss at the end of the experiment.

The theory behind this prepaid mechanism is founded in the mental accounting theory [42]. It is assumed that due to mental accounting, subjects considered the experimental money as their own wealth when receiving it two weeks prior to the session at which they made decisions. This induced a shift in the reference point for the decision in the DG afterwards. The status quo was now the money they already owned. Hence, when deciding about the EUR −10 and the necessity of paying it back, they were in a loss frame [26].

For ethical and practical reasons, it is not possible to take money from subjects for participating in scientific experiments. The prepaid mechanism is a practical solution to this problem, because overall, the subjects receive money for participating in the experiment, but in the session in which the game is played, they still decide on a loss. There are other mechanisms to induce losses, such as decisions over sports activities [43], waiting time [44,45], or even physical pain [46]. The problem with these methods is that they are not directly comparable to gains. The advantage of the prepaid mechanism consists of the fact that a decision about a monetary loss of EUR −10 is directly comparable to the decision about a gain of EUR 10.

We replicated these experiments in an online setting by using zTree unleashed [47] and web conference software. These online experiments were conducted in summer 2021. We used the same instructions and experimental files as in the laboratory experiments in 2018. The only difference was that the game instructions in laboratory conditions were given to the subjects on paper, whereas in online conditions, the instructions were included in the experimental files. In addition, payments in the laboratory condition were given in cash, while payments in the online condition were conducted by bank transfer.

We compared the results of this replication with the original experiments from 2018 and did not find a significant difference in behavior between online and offline conditions [40]. Therefore, we pooled the data of laboratory and online settings and used the DG data from both experiments as a control condition without the punishment mechanism. From the replication experiment, of course, we only used DG data in which the DG was played first.

### 2.2. Design of Punishment Experiments

We ran the TPP-DG experiments in summer and autumn 2021 and another series in the loss domain in autumn 2022. In the gains domain, the experiment was a standard TPP-DG, like in Fehr and Fischbacher [16]. Subjects were told that they were assigned to groups of three players. The groups were randomly matched, and the players’ roles were assigned randomly. For reasons of comparability, we used the same game instructions as in the DG in the control experiments and called this the first stage of the game. The dictator (Player 1) had to divide an amount of EUR 10 between themselves and the receiver (Player 2) in increments of 50 cents. For economic reasons of sample size, we did not play with receivers in the first stage. The instructions for this first stage were exactly the same as in the control experiments without punishment.

In the second stage, the dictator’s decision was shown to the punisher (Player 3), who had to decide whether to punish Player 1. The punishment mechanism was kept more simple than in other experiments [16]. If Player 3 decided to punish Player 1 for unfair behavior, then a specific amount of money was deduced from Player 1′s outcome. This amount of money varied in three conditions. In the weak punishment condition, the dictator received punishment via a deduction of EUR 1 from their outcome from the first stage. In the moderate punishment condition, the dictator had EUR 2 deducted from her outcome. In the strong punishment condition, the payoff of Player 1 was deduced by EUR 6. The punishment conditions were randomly assigned to experimental sessions.

Similar to other experiments [15,16], the decision to punish was costly. If Player 3 decided to punish, they paid EUR 2 for this decision in every treatment condition. Hence, in the weak punishment condition, Player 3 paid EUR 2 to punish, and Player 1 was penalized EUR 1. In the moderate condition, Player 3 paid EUR 2 to punish, and Player 1 was penalized EUR 2. In the strong condition, Player 3 paid EUR 2 to punish, and Player 1′s payoff was reduced by EUR 6.

The punisher received an endowment of EUR 10 in advance, from which they could pay the punishment cost of EUR 2. In the stage involving Player 3, Player 1′s decision from the first stage was shown, together with the resulting outcomes for Players 1 and 2. Below this, there were two checkmarks, one for “Penalize Player 1” and one for “Do not punish”. In addition, the consequence of punishing Player 1 was explained again. If Player 3 decided to punish, EUR 1 (or EUR 2 or EUR 6) was deducted from Player 1′s outcome, and Player 3 had EUR 2 deducted (see the Appendix A for all game instructions).

An advantage of this punishment mechanism is that its simplicity made it easy to implement various strengths of punishment by varying the amount deducted from Player 1′s payoff. This simplicity could also be seen as a limitation, because it is not possible to measure the strength of the desire to punish for Player 3, as in other studies. However, because our research interest was the fairness behavior of Player 1, this limitation was accepted.

For the negative TPP-DG experiments, we used an analogous design, as in the control experiments described above. In a first session, all subjects filled out a Big Five questionnaire and received a payment of EUR 15, which included the EUR 5 participation fee. They agreed to a statement saying that they would participate in the second session and that the given endowment might decrease at the second appointment. In the second session, two weeks later, subjects played a negative TPP-DG in which the dictator had to allocate a loss of EUR −10 between themselves and the receiver. Again, participants were told that the result of this allocation had to be paid back after the experiment. In the second stage of this game, Player 3 also had to decide whether to punish Player 1. For punishment, Player 3 had to pay back EUR 2 after the experiment. The effect of punishment on Player 1 varied between EUR 1, EUR 2, and EUR 6, like in the gains treatment. There were only two differences in the game description between the gains and loss treatment. First, in the first stage, Player 1 had to decide on a loss instead of a gain. Second, the penalty for Player 1 in the second stage was described as an “additional loss” instead of a deduction from the payoff. All other instructions were kept identical to the gains treatment.

All in all, we used a 2 × 4 design, as shown in Table 1. There was a control condition without punishment for gains (T1) and for losses (T2). The number in parentheses indicates the case number of dictators and receivers in the DG. An odd number is possible, because for sessions with an odd number of participants, there was one randomly chosen dictator decision without a corresponding receiver. In the TPP-DG, there were three conditions varying the strength of the punishment (weak, moderate and strong). These three treatments were applied to the gains and loss domain, resulting in six punishment treatments (T3–T8). All games were played in one-shot, and subjects participated in only one treatment. In the punishment conditions, we did not play with receivers in the DG, so that the number in parentheses shows the case number for dictators and punishers of the respective treatment. Like in the control treatment, for an odd number of participants in a session, there was one randomly chosen dictator decision without a punishment decision (these dictators never received punishment).

### 2.3. Procedure and Experimental Sample

The experiments were conducted in Leipzig and Magdeburg. In total, we had *n* = 672 participants. The mean age of our subjects was 27, with a standard deviation of 6.5 years. The majority of our participants were female (62.2%), and most (88.5%) had a student background. In Table 2, some demographic characteristics are shown for the gains and loss treatments separately. As can be seen, for age, gender, and national background, the subjects’ characteristics are similar in the gains and loss treatments. Only for the student background is there a difference, with fewer student participants in the gains treatments.

The subjects were invited from the subject pools of the Magdeburg Experimental Lab for economics (MaXLab) and the Leipzig Experimental Lab for social sciences (LEx). Both labs work with the recruiting system hroot [48]. This means that both labs run a website where people, usually students, can register for experiments. They agree to some basic rules of experimentation such as arriving on time to the experimental sessions or being paid for participation. From these subject pools, possible subjects are assigned randomly to experimental sessions and receive an invitation to participate. With enrollment in a session, they consent to take part in the experiment. In the loss treatments, every experiment consisted of two sessions. When subjects decided to take part in the experiment, they always enrolled in both sessions.

When an experimental session started, we gave some general instructions to the participants. These instructions included rules of conduct, particularly that communication with other participants was not allowed. Furthermore, the payment procedure was explained in these general instructions (refer to the Appendix A for general instructions).

In the control experiments from 2018, subjects were invited to computer pools in Magdeburg and Leipzig. There were screen walls at the experimental seats in both labs, so that participants could not see each other’s screens. This guaranteed single-blind anonymity when subjects were matched into groups during the experiment.

The online experiments—replication of the control experiments and punishment conditions—were conducted via a zTree unleashed server [47]. The subjects were invited to an online conference room. Cameras were turned off, and all subjects received an individual participation code in their invitation to use as a name in the conference tool. This procedure guaranteed single-blind anonymity. The names of the experimenters were fully written out in the web conference environment. The subjects were instructed not to use the microphone but to ask questions via private chat so as not to influence the other participants’ behavior. We provided the link to the experimental software via public chat. The experimental files opened on the participants’ devices.

We used zTree [49] as experimental software for all our experiments. Participants were randomly assigned to the groups and roles in the game, as explained in the instructions. In all games, subjects first read the instructions, and afterwards, they were told which role they had. In all treatments, subjects first made their decisions in the games and filled out questionnaires afterwards. The repayments of losses resulting from the decisions in the loss treatments were conducted after the experiment. In the laboratory experiments, these were paid in cash. Payments in all online sessions were conducted by bank transfer.

## 3. Results

### 3.1. Elicitation of the Fairness Norm

Before we test our hypotheses on the effects of punishment mechanisms on fairness in the next subsection, we show results concerning the question whether there is a norm of fairness involved in the TPP-DG. We used a norm elicitation method adapted from Bicchieri und Chavez [50,51], asking participants in our TPP-DG experiments for first- and second-order beliefs about fairness. We showed a list of the possible integer outcomes for Players 1 and 2, dividing the EUR 10 for the gains domain and EUR −10 for the losses domain. In the first question, we asked subjects to check outcomes they considered fair for Players 1 and 2. Second, we asked them to rate which of the possible integer outcomes would be considered fair by the majority of the participants. The elicitation was not incentivized.

In Table 3, we show fairness beliefs for selfish, fair, and altruistic dictator’s demands. The dictator’s demand is the proportion of the EUR 10 the dictator keeps in the gains domain. In the loss domain, the dictator’s demand is the proportion of the loss of EUR −10 that is transferred to the receiver. Clearly, the equal split is evaluated as fair for the dictator and receiver. In the gains domain, equal division is rated as fair for both players by more than 89% of the participants. In the loss domain, this proportion is nearly 94%. Small deviations from the equal split in both directions are still rated as fair for both players by around every fifth subject (in the case of a dictator’s demand of 60% around 30% in the gains domain). Only a few subjects rated different divisions of the amount to be shared as fair for both players, in most cases less than 7% of the respondents. Interestingly, in the gains domain, the division (7; 3) is rated as fair for both players by more than 11% of the participants, whereas the corresponding division (−3; −7) in loss domain is rated fair for both players by less than 3%. This difference is significant (Chi^2^ = 12.22; *p* < 0.001). Additionally, participants rated the equal split as fair for both players slightly more commonly in the loss domain, but the difference is not significant on the 5% level (Chi^2^ = 2.98; *p* = 0.084). It seems as if the fairness norm of equal sharing is slightly stronger in the loss domain compared to the DG in gains.

According to Bicchieri [2], for a social norm to exist, it is not just a matter of what people think is right or wrong in a certain situation; it is important what social expectations they have about what others might judge to be right or wrong. These second-order beliefs about what is fair in the underlying DG are shown in Table 4. The proportions of fairness ratings for dictators’ demands are nearly the same as in the first-order beliefs. The only difference is in the gains domain, where a higher proportion of respondents believe that the majority of participants would judge slight deviations from the equal split as fair, compared to first-order beliefs. Again, we find higher proportions of second-order fairness beliefs for the gains domain compared to losses. This indicates that subjects expect others to have a slightly stronger norm of fairness when dividing losses compared to gains.

These data cast no doubt that there is a social norm of fairness in the TPP-DG telling the dictator to play the equal split. As with many social norms, small deviations from the rule—in this case, plus/minus 10%—are more or less allowed. Bicchieri [52] (p. 36) states, “It is often the case that norms are not ‘all or nothing’ affairs. Fair divisions, for example, may include a 60–40% share as acceptable”. Now that we know that from an idealistic perspective, there is a social norm of fair giving in the DG, we investigate fairness behavior in punishment conditions.

### 3.2. Effects of Punishment on Fairness Behavior

To analyse the effect of punishment mechanisms on fairness in the DG, we compare the average dictator’s demands for every treatment. In the gains domain, the dictator’s demand is the proportion of the EUR 10 they keep. In the loss domain, the dictator’s demand is the proportion of the loss of EUR −10 that is transferred to the receiver. The corresponding results are shown in Figure 1. The first two bars on the left show the average demand from the control experiments without punishment. These data stem from laboratory and online experiments. The other bars show mean values for the punishment treatments. From left to right, the weak punishment treatments T3 and T4, the moderate treatments T5 and T6, and strong punishment treatments T7 and T8 are presented.

Comparing the values of the control treatment without a punishment mechanism and the figures concerning strong punishment shows an effect of punishment on fairness behavior. For gains as much as for losses, the dictator’s mean demand decreases when the decision is potentially sanctioned by a third party. However, the decrease in means is statistically significant only for the loss domain (t = 1.7; *p*[one-tail] = 0.047). In the gains domain, the population is already quite fair (mean = 63.98), which means that the introduction of a TPP mechanism cannot make a huge influence, because the level of fairness is capped downward.

Figure 1 also shows that the strength of the punishment makes a difference in the effect on fairness behavior. In the gains domain, the dictator’s mean demand is even higher for weak and moderate punishment mechanisms. In the loss domain, there is no difference between a moderate and a strong punishment mechanism. Furthermore, if we test differences between weak and strong punishment mechanisms, only the difference in the gains domain (69.3 versus 60.4) is statistically significant (t = 1.94; *p*[one-tail] = 0.028). Interestingly, for the weak punishment condition (2:1) compared to the control conditions, there is an increase in demand from 64.0 to 69.3 in the gains domain. However, this difference is not statistically significant (t = −1.14; *p*[two-tail] = 0.259).

Next, we test these findings with multiple OLS regression. Table 5 shows regression coefficients for the main effects of our treatments on dictators’ demands. Overall, for pooled data of punishment treatments with the control treatment, there is no effect of the loss condition against gains (M-Loss). In the punishment mechanism treatments (M-Punish), there is an effect on fairness only for the strong condition. Dictators in the strong punishment condition demanded 55 cents less on average compared to the control condition without a punishment mechanism. This effect remains robust if we control for the loss condition (M-Full). However, the coefficient is only weakly significant at 10%.

All in all, introducing a TPP mechanism has a decreasing effect on the dictator’s demands (cf. H1), but only if the mechanism is strong enough. The effect on fairness varies with the strength of the punishment mechanism (cf. H2). Furthermore, in the gains domain that already had a high level of fairness in the control condition, the introduction of a weak punishment mechanism has a deteriorating effect on fairness (cf. H3). However, only weak statistical significance exists for all these differences.

The preceding section showed the social norm of equal sharing in the TPP-DG. Next, we investigate the impact of the punishment mechanism on norm conformity. As seen in Figure 2, the strong punishment mechanism increases the proportion of equal split decisions compared to the control treatment without any punishment. The loss domain exhibits this increase even for all punishment treatments. Yet, the increase from around 40% to more than 50% is not statistically significant according to proportion tests (z = −1.216; *p*[one-tail] = 0.112 for gains; z = −1.128; *p*[one-tail] = 0.130 for losses).

If we stick to Bicchieri’s [52] dictum, namely that deviations of ±10% from equal sharing might still be seen as adherence to the norm and, thus, fair sharing, the data can be interpreted in this way. Therefore, we define all divisions where the dictator’s demand is between 40% and 60% of the pie as fair. These figures are shown in Figure 3. The loss domain shows an increase in the proportions of fair splits for all three punishment conditions compared to the control treatment without punishment possibility. In the gains domain, this pattern is different. The strong punishment exhibits an increase in fair splits from 65% to 76%, but for weak and moderate punishment conditions, norm conformity decreases. Namely, the proportion of fair splits decreases from 65% to 55% for weak punishment and 57% for moderate punishment. For gains, Hypothesis 3 seems to hold: Introducing a weak punishment mechanism makes dictators even less fair. However again, these differences are not statistically significant.

This section provided evidence for all three stated hypotheses, H1–H3. Introducing a punishment mechanism shows an increasing effect on fairness (in means, equal splits and fair splits), but only for the strong punishment mechanism. Furthermore, in the gains domain introducing a weak punishment mechanism even decreases fairness, measured with mean demands, as much as with equal and fair splits. Nevertheless, most of these differences are not statistically significant; therefore, this evidence is rather weak.

### 3.3. Interaction Effect of Punishment in the Loss Domain

A deeper look at the differences of the effect of a TPP in DG between gains and losses provides insight on two contradicting hypotheses. Either the introduction of a punishment mechanism does not affect the difference in fairness behavior between gains and losses (H4), which would mean a pattern from control treatment with higher demands in the loss domain for all punishment conditions, or, as the competing hypothesis states, there is an interaction effect of the punishment mechanism with losses in a way that the introduction of a TPP mechanism shows a stronger effect on fairness when sharing losses compared to gains (H5).

The preceding section already showed the corresponding evidence. The loss domain has an effect of punishment on mean demands with a stronger punishment mechanism showing a stronger effect on fairness (Figure 1). Measuring norm conformity with the proportion of equal splits (Figure 2) or fair splits (Figure 3) in the loss domain reveals that all punishments mechanisms show more or less the same impact on conformity to a fairness norm independent of its strength. This does not hold for the gains domain. There, only the strong punishment has an increasing effect on norm conformity. It decreases the mean demands and increases fair splits. However, in contrast to the loss domain in the gains domain, a weak and moderate punishment mechanism leads to a deteriorating effect on norm conformity. From this evidence, hypothesis H4 must be rejected and hypothesis H5 can be accepted. There is an interaction effect of a TPP mechanism with the loss domain.

We also test this finding for interaction effects between punishment and the loss condition with OLS regression analysis (Table 6).

For the weak punishment condition, again, there is no significant effect on the dictator’s demands. Under the strong punishment condition, the coefficient becomes insignificant when adding the interaction term between punishment and loss condition. Interestingly, dictators who expect moderate punishment become (on 10% level) significantly fairer in the loss domain compared to the gains and punishment condition. This suggests that sanctioning mechanisms have a stronger effect on conformity to a fairness norm when losses are divided than when gains are at stake.

### 3.4. Punishment Behavior

Overall, there are 232 punisher decisions, of which 35 (15.1%) have been punishments. Most of these punishments (82.9%) were executed on unfair demands. Overall, one-third of all unfair demands (33.7%) have been punished.

As Table 7 shows, most punishments took place with the moderate punishment. There, 20.5% of all demands were punished. With both other punishment mechanisms, only 14.3% (weak) and 10.1% (strong) of the dictator’s decision received punishment. In every condition, there was a low number (between 1 and 3) of fair demands that received punishment. Interestingly, with the strong punishment mechanism, the proportion of the punished unfair demands was low (26.1%) compared to the weak and moderate mechanism. In general, there have been fewer unfair demands under the strong punishment condition, but from these unfair demands, even fewer received punishment. This might be an effect of a do-no-harm norm because punishment would have harmed the dictator with EUR 6. With the strong punishment mechanism, dictators hesitated to make unfair demands, but also punishers hesitated to punish.

From the behavioral perspective on social norms, we interpret this result as evidence that in the DG with third-party punishment, an activated social norm prescribes the dictator to give a fair share. There was a high proportion of punishments for unfair shares. In contrast to this, fair shares nearly never received punishment. If this socially shared rule that demands a dictator in a DG to give a share of 40% to 60% is transgressed, there is a high chance that the transgressor will be sanctioned.

Due to case numbers, it is hard to make distinct interpretations for the case on the gains versus the loss domain. Nevertheless, punishment seems to be more present in the loss domain.

## 4. Discussion

We conducted experiments on the effect of TPP on sharing losses in a negative DG and compared these results to standard TPP-DG on gains. Mainly, we are interested in the effect of the punishment mechanism on fairness in the loss domain compared to standard games on gains. Furthermore, we vary the strength of the punishment mechanism by its impact on the dictator’s outcome. The costs of punishment for the third party were kept constant. We compared all 2 × 3 punishment conditions with evidence from negative and positive DGs without punishment.

For the effect of the punishment mechanism on the fairness norm, we find that the strength of the punishment mechanism matters. If the punishment mechanism is too weak, and especially if its costs do not outweigh the benefits from being unfair, the punishment threat hardly works. Then, it might even have a detrimental effect on norm conformity [53]. The latter holds especially for the gains domain. In the loss domain in contrast, punishment mechanisms generally seem to be more effective in maintaining adherence to the fairness norm.

Regarding punishers’ sanctioning behavior, we found that punishment of unfair demands took place predominantly at the moderate punishment condition. The weak punishment mechanism might have been harmless in the eyes of many punishers (and dictators). For the strong punishment mechanism in contrast, there might have been a norm of do no harm.

Nevertheless, our study has some limitations. First, our sample consists of students from various fields of study and they were recruited from two subject pools located at two German Universities (Leipzig and Magdeburg). These aspects are considered to limit the generalizability of our results, especially in light of the results of studies comparing fairness behavior of students versus nonstudents. These studies regularly find that students decide in a more self-interested way [54,55]. It would be interesting to replicate the experiments with a more diverse sample in other countries. Second, overall, the statistical evidence for our results is rather weak. Though we have close to 50 decisions per condition, this is still due to the rather low case numbers that are typical for experimental designs. More importantly, our research population is already quite fair. On average, dictators in DG experiments demand around 72% [14]. However, in our control condition, the average demand is lower, with 64% for gains and 70% for losses. Consequently, there is less space left for the effect of sanctioning on fairness behavior. For that reason, further research is required.

With regard to the question whether there is a social norm of fair sharing in the DG [7], our results can be interpreted as follows: From the behavioural perspective on norms [4], the most probable interpretation of the data is that such a norm exists. First, the dictator’s mode demand, and typically also the mean demand, was the fair share in the DG. This holds even more for the loss domain, and a situation in which TPP is possible. Also, norm deviance here undoubtedly is sanctioned, and consequently, the punishment threat, if it is strong enough, has a positive effect on norm conformity. From the idealistic view [2], the evidence for a social norm is even stronger. First, subjects overwhelmingly state that they have a fairness norm of equal sharing for themselves. Second and more important, they think even more that the others also have a similar norm. However, for gains, these first- and second-order beliefs on the fairness norm are not completely congruent; while in the loss domain, both beliefs closely coincide. The latter is especially the crucial requirement for the existence of a social norm.

Also, our assumptions on the loss domain were confirmed. There is a difference between decisions in the loss and gains domain with strategic decisions that generate real outcomes. This makes it worthy to research it. As we found in our previous research and stated in the prospect theory [26,39], “Losses loom larger than gains” [39] (p. 279). When it comes to losses, subjects become more rational and straightforward with their words as much as with their deeds. There seems to be no manna economy [56] in the loss domain. The wiggling [36] and wavering then comes to an end when resources are scarce and things become serious.

Our study contributes to the literature on a theoretical and empirical level. We showed evidence that, at least in the DG with third-party punishment, a social norm of fair sharing guides a dictator’s behavior. We also showed that punishment increases conformity to that norm, but only when the punishment mechanism is strong enough. This finding confirms norm theory [2,28]. Most importantly, our results add some first exploratory findings to the field of cooperation and fairness behavior in the loss domain. Especially, we found that in the loss domain sanctioning seems to have a stronger impact on conformity to a fairness norm compared to the gains domain. Further research should be conducted on the question of whether there generally is an interaction effect of punishment and losses that increases conformity to norms of fairness or cooperation. If this is the case, this would have strong implications for real-world situations dealing with losses (e.g., pandemic crisis or climate crisis) because in such situations, sanctioning would increase cooperative behavior also with weak mechanisms.

## Figures and Tables

**Figure 1 behavsci-14-00039-f001:**
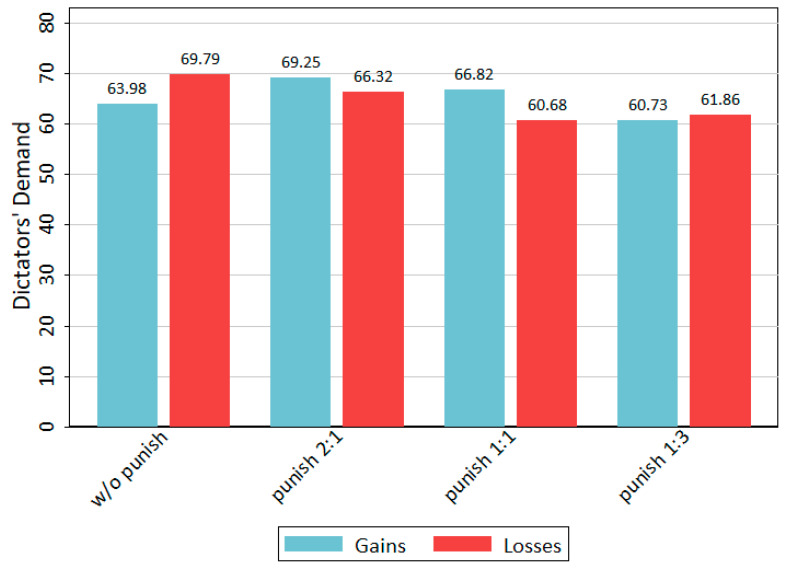
Mean dictator’s demands per treatment.

**Figure 2 behavsci-14-00039-f002:**
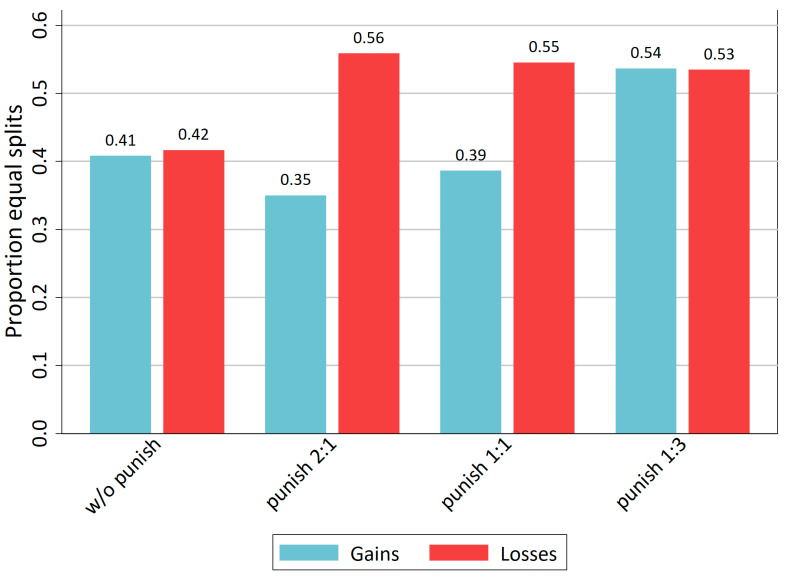
Proportion of equal splits per treatment.

**Figure 3 behavsci-14-00039-f003:**
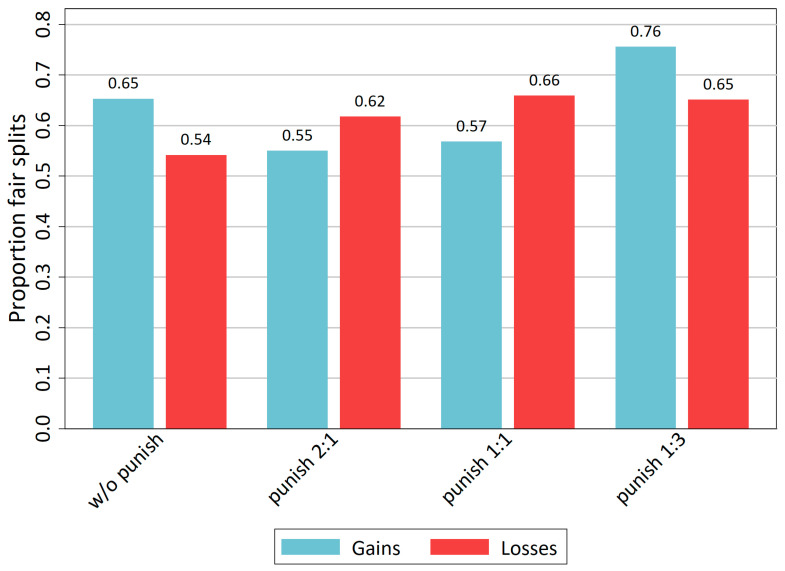
Proportion of fair splits per treatment.

**Table 1 behavsci-14-00039-t001:** Treatment overview.

Punishment Condition	Gains	Losses
Without punishment (Control)	T1 (*n* = 96)	T2 (*n* = 93)
Lever 2:1 (Weak)	T3 (*n* = 78)	T4 (*n* = 68)
Lever 1:1 (Moderate)	T5 (*n* = 86)	T6 (*n* = 87)
Lever 1:3 (Strong)	T7 (*n* = 80)	T8 (*n* = 84)

Comment: “Lever” shows the ratio between punishment costs for the punisher and the punishment impact on the dictator. The number in parentheses shows the number of subjects participating in the treatment. We have *n*/2 dictator’s decisions and *n*/2 punisher’s decisions. In case of odd numbers, we have one (randomized) dictator’s decision without a punishment decision; dictators never received punishment in this case.

**Table 2 behavsci-14-00039-t002:** Demographics.

Characteristic	Gains	Losses
Age	27.2	26.7
Female	63.2%	63.9%
Studies	80.5%	98.5%
German	93.4%	94.4%

Comment: For age, the average value is shown. For the other variables, the percentage is shown where the particular characteristic is true.

**Table 3 behavsci-14-00039-t003:** First-order fairness beliefs.

	Selfish		Fair			Altruistic	
Dictator’s demand *d*	d≥80%	d=70%	d=60%	d=50%	d=40%	d=30%	d≤20%
% in gains (*n* = 244)	≤5.74%	11.07%	29.51%	89.34%	20.49%	6.56%	≤3.69%
% in losses (*n* = 239)	≤2.09%	2.93%	19.25%	93.72%	18.83%	4.18%	≤3.77%

Note: The table shows the proportions of positive answers to the question “Which of the following allocations do you consider fair for Player 1 and Player 2?” The question showed all possible integer divisions in the form (x; 10 − x) for gains and (x; −10 − x) for the loss domain, where x is the share for Player 1.

**Table 4 behavsci-14-00039-t004:** Second-order fairness beliefs.

	Selfish		Fair			Altruistic	
Dictator’s demand *d*	d≥80%	d=70%	d=60%	d=50%	d=40%	d=30%	d≤20%
% in gains (*n* = 244)	≤2.87%	8.20%	36.89%	86.48%	24.59%	3.28%	≤0.82%
% in losses (*n* = 239)	≤1.26%	5.02%	20.92%	93.31%	21.34%	5.44%	≤2.51%

Note: The table shows the proportions of positive answers to the question “Which of the following allocations are considered fair for Player 1 and Player 2 by the majority of the participants in this experiment?” The question showed all possible integer divisions in the form (x; 10 − x) for gains and (x; −10 − x) for the loss domain, where x is the share for Player 1.

**Table 5 behavsci-14-00039-t005:** Regression of main effects of dictator’s demand on losses and punishment strength.

Variable	M-Loss	M-Punish	M-Full
Loss domain (=1 loss)	−0.44 (0.850)		−0.28 (0.903)
Weak punishment (=1)		1.05 (0.751)	1.04 (0.754)
Moderate punishment (=1)		−3.11 (0.326)	−3.10 (0.327)
Strong punishment (=1)		−5.55 (0.084)	−5.54 (0.084)
Constant	65.14 (0.000)	66.86 (0.000)	67.00 (0.000)
N	343	343	343
Adj. R^2^	−0.003	0.005	0.003

Note: Table shows OLS regression coefficients where the dependent variable is the dictator’s demand; figures in parentheses show *p*-values. The reference treatment for M-Punish is the control treatment without a punishment mechanism, and for M-Full, the gains treatment without a punishment mechanism.

**Table 6 behavsci-14-00039-t006:** Regression of interaction effects of dictator’s demand on losses and punishment strength.

Variable	M-Weak	M-Moderate	M-Strong
Loss treatment (=1 Loss)	5.81 (0.186)	5.81 (0.189)	5.81 (0.180)
Weak (=1 weak punishment)	5.27 (0.253)		
Loss × Weak	−8.74 (0.191)		
Moderate (=1 moderate *p*.)		2.84 (0.529)	
Loss × Moderate		−11.95 (0.063)	
Strong (=1 strong *p*.)			−3.25 (0.472)
Loss × Strong			−4.68 (0.461)
Constant	63.98 (0.000)	63.98 (0.000)	63.98 (0.000)
N	171	185	181
Adj. R^2^	−0.005	0.008	0.011

Note: Table shows OLS regression coefficients where the dependent variable is the dictator’s demand; figures in parentheses show *p*-values. The reference treatment is always the control condition without a punishment mechanism. M-Weak compares weak punishment against control, M-Moderate compares moderate punishment against control, and M-Strong compares strong punishment against control.

**Table 7 behavsci-14-00039-t007:** Punishment behavior.

	Weak		Moderate		Strong		
	Gains	Losses	Gains	Losses	Gains	Losses	Total
Fair	0	1	1	2	0	2	6
Unfair	6	3	8	6	2	4	29
Total	6	4	9	8	2	6	35
Proportion Total	10/70 or 14.3%		17/83 or 20.5%		8/79 or 10.1%		35/232
Proportion Unfair	9/30 or 30.0%		14/33 or 42.4%		6/23 or 26.1%		29/86

Note: The table shows the number of punishments for fair and unfair splits. Fair is defined as a dictator’s demand between 40% and 60%, including boundary values. Proportion total shows the number/proportion of punishments for all splits per punishment condition. Proportion unfair shows the number/proportion of punishments for unfair splits per punishment condition.

## Data Availability

The data presented in this study are available in Appendix A.

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
