# Peer review of "Enforcement of Fairness Norms by Punishment: A Comparison of Gains and Losses"

_behavsci, 2024, doi:10.3390/bs14010039_

Round 1

Reviewer 1 Report

Comments and Suggestions for Authors

Thank you for giving me an opportunity to review this paper. This paper conducts online experiments with a large sample size of 672 participants, providing a robust dataset for analysis.

The study explores the role of loss and punishment threats on conformity to the fairness norm, shedding light on decision-making in the loss domain.

The findings suggest that individuals are more rational and straightforward in their decisions and perceptions of fairness in the loss domain compared to gains.

The study reveals that dictators in the loss domain are more responsive to punishment threats, reducing their demands when faced with strong punishment.

Weakness:

·        The statistical evidence overall is weak, indicating that further research is needed to strengthen the findings.

·        The paper does not provide detailed information on the experimental design and methodology, limiting the ability to assess the study's internal validity. 

Areas of Improvement:

·        The paper could benefit from providing more information on the demographic characteristics of the participants, such as age and gender, to assess the generalizability of the findings. 

·        Including a control condition with no punishment threat in the loss domain would allow for a more comprehensive comparison between gains and losses. 

Reviewer 2 Report

Comments and Suggestions for Authors

This manuscript reports on an online experiment aimed at exploring the role of loss and punishment threats on conformity to the fairness norm in third-party punishment dictator games. The study involved 672 participants who were randomly assigned to different treatment conditions, including a control condition with gains and no punishment, and several conditions with losses and varying levels of punishment threats. The results suggest that, overall, participants are more rational and straightforward in their decisions and responses when facing losses than when facing gains. They tend to adhere more strongly to the fairness norm in the loss domain and perceive that others do the same. Additionally, the study found that dictators in the loss domain are more responsive to punishment threats and are more likely to reduce their demands when faced with strong punishment threats, as judged from a rational perspective.

The main contribution of this study is its focus on decision-making in the loss domain and the role of punishment threats in promoting conformity to the fairness norm. The results suggest that losses may lead to more rational and fair decision-making, and that punishment threats can effectively deter selfish behavior in the loss domain. The study provides valuable insights into the psychology of decision-making in the context of losses and highlights the importance of considering the role of punishment threats in promoting cooperation and fairness in social interactions.

The manuscript is written in standard academic English and presents the research question, methodology, results, and conclusions in a clear and concise manner. It also discusses the implications of the findings and suggests directions for future research.

Below are a few questions that I would like to receive answers from the authors.

What were the different punishment conditions implemented in the TPP-DG experiments in the gains and losses domains?

How were the roles and groups assigned to the participants in the experiments?

What were the specific instructions given to the dictator, receiver, and punisher in the TPP-DG game?

The authors can discuss the impact of punishment intensity and economic intensity on fairness, which seems to be an interesting point, as previous theoretical research has mentioned its (see Applied Mathematics and Computation 425 (2022): 127069).

How did the punishment impact on the dictator vary across the different treatments?

Can you provide an overview of the treatment conditions and the number of subjects participating in each condition?

What was the procedure followed for recruiting participants for the experiments conducted in Leipzig and Magdeburg?

How was anonymity ensured in the computer pool experiments, and what software was used for conducting the experiments?

What were the specific instructions given to the subjects regarding their behavior during the experiments?

How many subjects participated in the experiments, and what were the demographic characteristics of the participants?

Reviewer 3 Report

Comments and Suggestions for Authors

Congratulations, your paper is very interesting.

Nevertheless, it is my opinion that your paper presents some major weakness, mostly due to the impossibility of readers to fully understand how your experiment was run.

I strongly recommend you to improve this point, adding some critical contents from your supplementary material.

Furthermore, you do need to show in detail the advantages and limitations of the proposed methodology. The possible contributions of your study must be outlined in detail..

Same for future research. For example, your last sentence "Therefore, we suggest that it is as fruitful for economics as much as necessary for the economy to do more research in the loss domain." is absolutely void. You need to be clear and detailed at all times.

Comments on the Quality of English Language

Nothing relevant to mention.

Round 2

Reviewer 3 Report

Comments and Suggestions for Authors

Thanks for trying to reply and justify your paper's weaknesses.

I agree with the other reviewers comments and I recognize that the authors made a serious effort to mitigate the detailed weaknesses pointed out.

It is also relevant to mention the clarity and transparency from the authors.

Anyway, somethings still make wonder, being the most relevant this Limitations quote:

- "First, as in most experimental studies our results 692 are only valid for a student sample and most of our participants were German."

Does this means that the sample is made essentially from students? (are the "subjects", students, after all? If so, then such should made very clear at start).

This is probably a problem of communication (writing), as not everybody on the sample is educated.

Comments on the Quality of English Language

Nothing relevant to mention

Author Response

Dear Reviewer 3,

thank you very much for your comments! You asked about the composition of our research sample and noted that we should communicate clearer that our sample is mainly students.

In subsection 2.3 “Procedure and experimental sample” it was already mentioned that most participants had a student background (line 428). To address your comment, we made two additions. We included the information about the student background of our participants in the abstract of the paper (line 15). In the discussion, we added the information that subjects were recruited from two German universities and also included some literature about fairness behaviour of students versus non-students (lines 682-687).

You also made a checkmark that minor editing of English language is required. Thus, we sent the manuscript to proofreading and improved the quality of the language.

Thank you very much.